# First Record of Spinal Deformity in the South American Silver Croaker *Plagioscion squamosissimus* (Eupercaria: Sciaenidae) in the Xingu River, Brazil

Luciano Fogaça de Assis Montag [1], Luiz Antônio Wanderley Peixoto [2], Lidia Brasil Seabra [1], Liziane Amaral Barbosa Gonçalves [3], Cleonice Maria Cardoso Lobato [4], Marina Barreira Mendonça [5], Tiago Octavio Begot [1], Erival Gonçalves Prata [1] and Tiago Magalhães da Silva Freitas [6,*]

1   Laboratório de Ecologia e Conservação, Universidade Federal do Pará, Rua Augusto Corrêa 01, Guamá, Belém 66075-110, PA, Brazil; lfamontag@gmail.com (L.F.d.A.M.); lidiabrasil95@gmail.com (L.B.S.); tbegot@gmail.com (T.O.B.); erival.gprata@gmail.com (E.G.P.)
2   Instituto de Estudos Costeiros de Bragança, Universidade Federal do Pará, Alameda Leandro Ribeiro, Aldeia, Bragança 68600-000, PA, Brazil; luizwp@yahoo.com.br
3   Laboratório de Histologia, Universidade Federal do Pará, Rua Augusto Corrêa 01, Guamá, Belém 66075-110, PA, Brazil; lizbarbipesca@gmail.com
4   Laboratório de Avaliação e Monitoramento da Biodiversidade, Programa de Pós-Graduação em Ecologia, Universidade Federal do Rio de Janeiro, Rua Moniz Aragão N° 360, Bloco 1, Ilha do Fundão, Cidade Universitária, Rio de Janeiro 21941-594, RJ, Brazil; lobatocmc@gmail.com
5   Laboratório de Ictiologia de Soure, Universidade Federal do Pará, Décima Terceira rua, s/n., Umirizal, Soure 68870-000, PA, Brazil; barreira.mm@gmail.com
6   Laboratório de Zoologia, Universidade Federal do Pará, Alameda IV, 3418, Parque Universitário, Breves 68800-000, PA, Brazil
*   Correspondence: freitastms@gmail.com

**Abstract:** Observations of skeletal malformations in fish in the wild are poorly documented and need to be investigated. Here we report the occurrence of body shortening in specimens of *Plagioscion squamosissimus* collected in the Volta Grande do Xingu, middle Xingu River region (Pará, Brazil), during a 12-month monitoring program (2021–2022). We observed morphological anomalies in nine individuals, of which two underwent radiographic analysis, recording the fusion and compression of vertebrae in different portions of the spine. The average percentage decrease in body length resulting from the deformity was 23.8%. This is the first record of malformation in this species.

**Keywords:** fish; spinal curvature; malformation; Amazon

**Key Contribution:** Our study reports the first record of skeletal deformity in *Plagioscion squamosissimus* found in the Xingu River.

## 1. Introduction

Deformities in fish have been recorded all over the world's watersheds, manifested in several ways, including fin, jaw, and spinal anomalies. Specifically, vertebral abnormalities, including flattening or fusion of vertebrae, can result in spinal curvature and/or shortening [1,2]. External evidence of deformities may be less apparent or absent when only a few vertebrae are affected. However, when multiple vertebrae are involved, observable morphological changes occur [3,4]. Often, alterations in internal bone structures precede external deformities, leading to delayed diagnosis and challenges in identifying the underlying cause of the malformation [2,5].

These malformations can arise from intrinsic factors, such as genetic mutations [3], and extrinsic factors, including environmental influences like climate changes and pollutants [6,7]. Notably, fish deformities have been utilized as valuable biomarkers to evaluate

environmental conditions [8,9], encompassing water quality parameters and the deposition of heavy metals [6,9].

In this context, the adverse consequences of mining activities and the construction of hydroelectric dams warrant careful consideration due to their detrimental effects on aquatic ecosystems [10,11]. A notable example is the Xingu River Basin, one of the largest drainage systems in the Amazon, which has experienced the construction of the highly controversial Belo Monte Dam, considered the most contentious hydropower project in South America [11]. Furthermore, the basin is also affected by illegal gold extraction [12]. These activities can potentially contribute to fish deformities [5,6], amplifying the environmental concerns associated with the region.

Given the above, this study marks the first documentation of osteological malformations, fusion, and shortening of the vertebral column in *Plagioscion squamosissimus* (Heckel, 1840), an endemic sciaenid fish from the Amazon Basin. Locally known as "pescada-branca" (or South American silver croaker), this species is important for local subsistence fishing [13], and in recent decades, it has been introduced into the major South American basins, mainly in reservoir lakes for commercial purposes [14].

## 2. Materials and Methods

Specimens were collected in the middle Xingu River region (Pará, Brazil) from December 2020 to November 2021, using gillnets with different mesh sizes. This section of the Xingu River Basin, commonly referred to as "Volta Grande do Xingu" (3°25′33.0″ S, 51°57′03.3″ W) (Figure 1), is of great ecological significance and recognized for its clear water and high flow, which runs over a bedrock, harboring a diverse flora and fauna [15]. However, this region is affected by the Belo Monte Dam, resulting in a 100 km stretch of reduced water flow and an 80% reduction in the overall flow. The biological effects of this environmental alteration are still far from being fully understood.

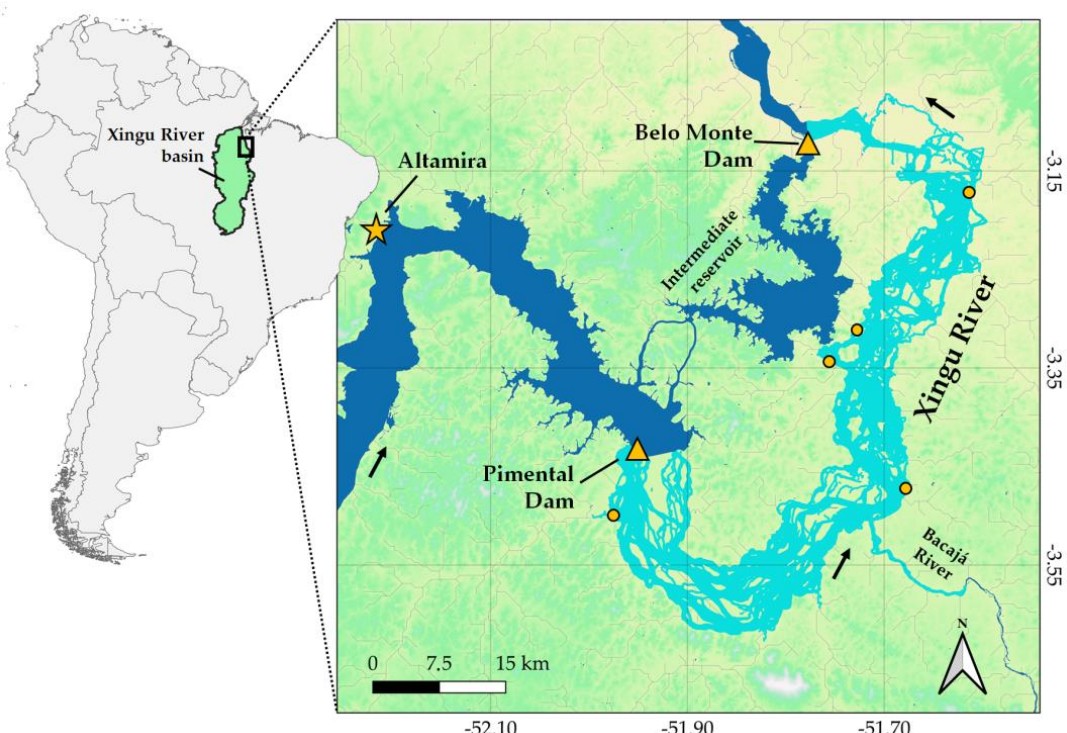

**Figure 1.** Location of the reduced flow section (highlighted in light blue) of the Xingu River in Pará State, Brazil, resulting from the construction of the Belo Monte Dam (including the Pimental Dam). The yellow circles indicate the sampling sites where specimens of *Plagioscion squamosissimus* with spine deformities were collected. The arrows indicate the direction of water flow.

After capture, the fish were anesthetized with benzocaine (0.1 g/L) and euthanized. We measured each individual's standard length (cm; SL; precision of 0.1 cm) and total mass (g; 0.1 g). A gonad fragment of these specimens was also obtained for sex and maturity determination through a histological routine. Collected specimens were preserved in 10% formalin for 48 h, later transferred to 70% ethanol, and stored in the Museu de Zoologia of the Universidade Federal do Pará (UFPA).

Here, we initially examined the external morphology of 483 collected individuals, out of which 9 exhibited spinal anomalies (body shortening). To confirm osteological changes, 2 specimens with deformities and 6 with regular body plans (Figure 2) were subjected to radiographic analysis, operated in the configuration of 100,000,000 mA and 65.00 kVp (WL:15771 WW:28071). The compound caudal centrum (preural 1 + ural 1) was counted as a single element [16].

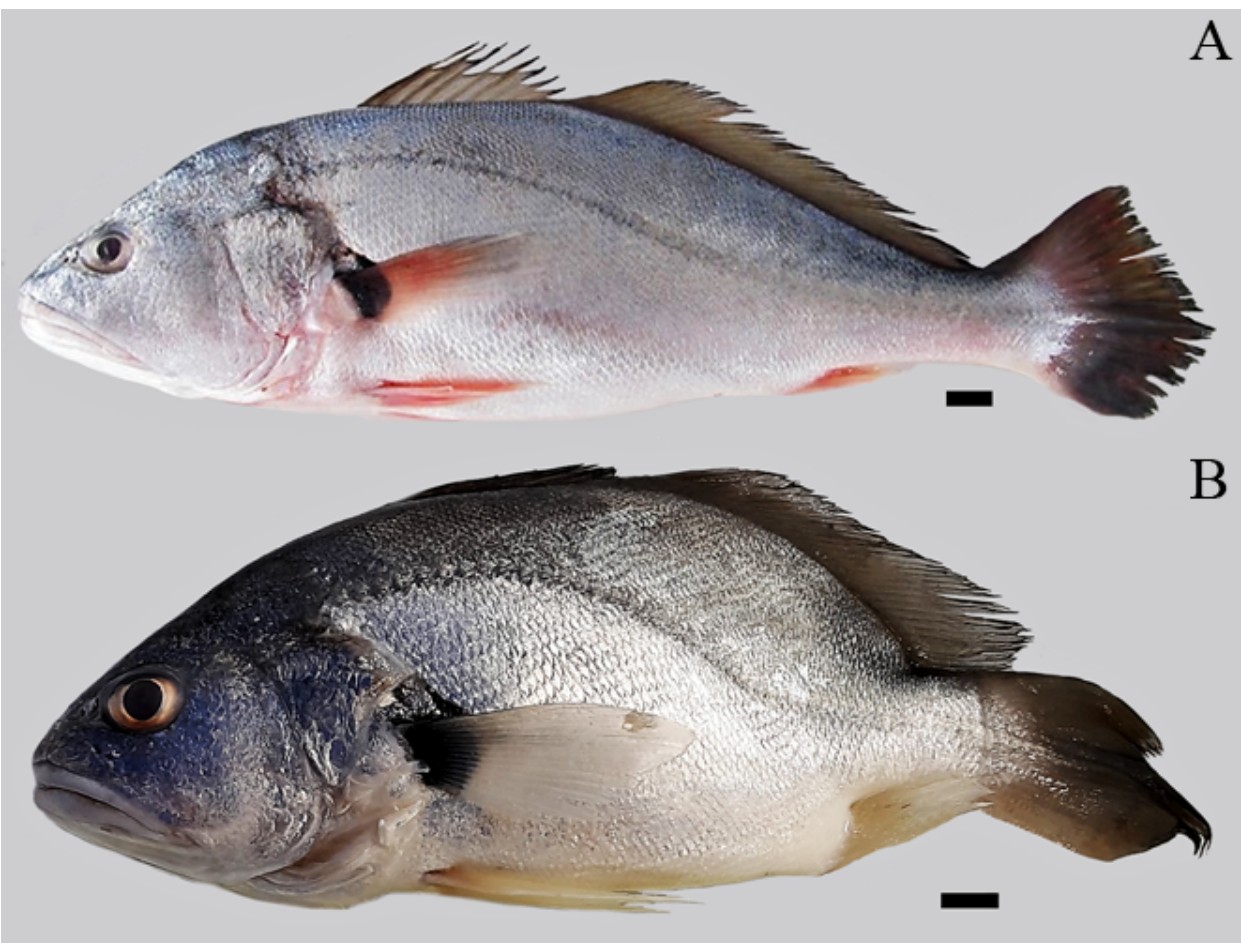

**Figure 2.** Specimens of *Plagioscion squamosissimus*; regular ((**A**) 25.4 cm SL) and with anomalies ((**B**) 22.5 cm SL). Bars = 1 cm. The coloration of individuals can vary due to the time that elapsed after collection and fixation.

## 3. Results

The specimens with a regular body plan exhibited a size range of 8.0 to 48.0 cm SL (mean = 22.9 cm; SD = 6.6 cm) and a weight range of 10.7 to 2205.0 g (292.9 ± 297.0 g), while anomalous individuals ranged from 9.3 to 22.5 cm SL (18.3 ± 4.1 cm) and 28.9 to 301.4 g (182.2 ± 100.7 g). Table 1 gives the size, sex, and maturity stage of the analyzed specimens with osteological anomalies. When comparing non-deformed fish with similar mass to anomalous specimens, we observed a rate of shortening of 23.8% (± 6.3%) (Table S1).

**Table 1.** Morphometry and sexing of *Plagioscion squamosissimus* specimens with deformities collected in the Volta Grande do Xingu, Xingu River (Pará, Brazil). n.d. = not determined.

| Ind | Standard Length (cm) | Mass (g) | Sex | Maturity Stage |
|---|---|---|---|---|
| Ind-1 | 9.3 | 28.9 | n.d. | - |
| Ind-2 | 15.1 | 69.0 | n.d. | - |
| Ind-3 | 16.4 | 122.0 | n.d. | - |
| Ind-4 | 19.5 | 165.0 | Female | Spawned |
| Ind-5 | 20.0 | 301.4 | Male | Maturing |
| Ind-6 | 20.1 | 276.0 | Female | Spawned |
| Ind-7 | 20.8 | 136.0 | Male | n.d. |
| Ind-8 | 21.0 | 261.0 | n.d. | - |
| Ind-9 | 22.5 | 280.2 | n.d. | - |

The rate of fish with spinal deformities was 1.9% (n = 9). Radiographed specimens showed the same numbers of vertebrae: 12 pre-caudal and 13 caudal vertebrae (including urostyle). However, we observed fusion and compression of vertebrae in the two deformed specimens. In one of them, we recorded three fusion points of vertebrae: (i) the 12th pre-caudal vertebra with the 1st caudal vertebra, (ii) the 3rd and 4th caudal vertebrae, and (iii) the 5th and 6th caudal vertebrae. Additionally, we recorded compression of all caudal vertebrae; a few pre-caudal centra were also compressed compared to normal fish (Figure 3: 10P, 11P, and 12P). In the other specimen, we observed only compression of pre-caudal vertebrae (5th, 9th, and 10th) and caudal vertebrae (11th and 12th).

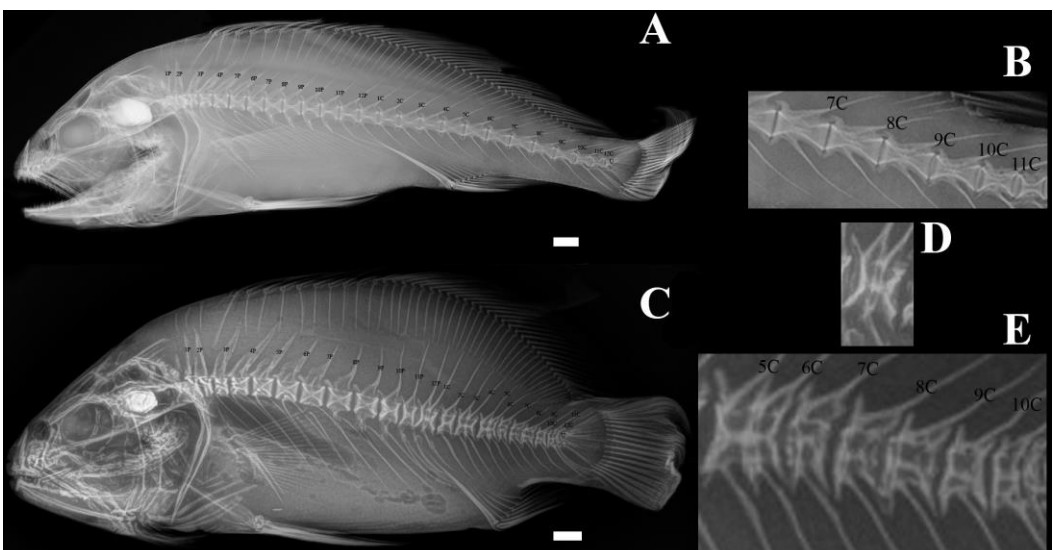

**Figure 3.** Radiographic analysis of specimens of *Plagioscion squamosissimus* ((**A**) regular body plan; (**B**) details of the caudal vertebrae; (**C**) anomalous condition; (**D**) vertebrae centra fusion; (**E**) details of the caudal vertebrae in the anomalous condition), collected in the Volta Grande do Xingu, Xingu River (Brazil). P = pre-caudal vertebrae; C = caudal vertebrae; U = urostyle. Bars = 1 cm.

## 4. Discussion

There is no reduction in the number of vertebrae in the deformed individuals of *P. squamosissimus* from the Xingu River (Amazon Basin), and the abnormal morphological pattern results from compression of the posterior caudal vertebrae and an apparent fusion of others. These data constitute the first record of osteological anomalies in *P. squamosissimus*, one of the most commercially important species in the Amazon [17]. Our hypothesis posits that observed deformities might have emerged during the early stages of growth, potentially driven by the interplay of two main factors. Firstly, the reduced water flow

due to the damming of the Xingu River may lead to increased mechanical loads on the fish's embryonic and larval stages. Secondly, the accumulation of heavy metals (e.g., mercury) and pollutants from illegal mining and agricultural activities (e.g., pesticides). Both factors have the potential to impact the development and the mineralization process of the notochord, which has been reported as a cause of vertebral compression and fusion in fish [5,18–20]. However, identifying the specific cause of skeletal deformities in natural fish populations is challenging, as they may be influenced by various interacting factors, including exposure to pathogens and nutritional aspects [3].

Further investigation is necessary to fully understand our findings due to the species' importance as a protein source for riverine communities and in the local fish markets [17]. Moreover, it is essential to enhance monitoring efforts targeting the natural fish populations in the middle Xingu River region to assess the ongoing impacts caused by the Belo Monte Dam. In addition to the necessity of increasing enforcement against illegal mining in the region, it is crucial to monitor the biological risks posed by this activity to fish populations.

**5. Conclusions**

Despite deformities, the analyzed individuals exhibited substantial growth, with several reaching or surpassing reproductive size. We emphasize the need for in-depth investigations into the possible causes of fish deformities in the Xingu River and the implementation of a long-term fish-monitoring program that includes *P. squamosissimus* populations and water quality parameters (e.g., water temperature, turbulence, and heavy metal concentration) to implement conservation measures and protect the local ichthyofauna and its environment.

**Supplementary Materials:** The following are available online at https://www.mdpi.com/article/10.3390/fishes8070363/s1, Table S1: Standard length (SL) between deformed and normal individuals of *Plagioscion squamosissimus* within the same body weight range, and their respective differences.

**Author Contributions:** L.F.d.A.M. and T.M.d.S.F. designed the study. L.A.W.P. conducted the radiographic analysis of the manuscript. L.F.d.A.M., L.A.W.P., L.B.S., L.A.B.G., C.M.C.L., M.B.M., T.O.B., E.G.P. and T.M.d.S.F. wrote the original draft. All the authors contributed to writing the manuscript, reviewing, and editing. All authors contributed to the article and approved the submitted version. All authors have read and agreed to the published version of the manuscript.

**Funding:** LFAM thanks the National Council for Scientific and Technological Development (CNPq) for a research productivity fellowship (Grants #302881/2022-0). This study was partially funded by the Coordination of Improvement of Higher Education Personnel—Brazil (CAPES)—Financial Code 001; We are grateful for funding from authors' grants EGP 88887.615449/2021-00, LBS 88887.615440/2021-00, and LAWP (FAPESP #2018/05084–1,CNPq #168395/2022-3, and FAPESPA #2023/158693). The authors also thank the Norte Energia S/A and Tractebel Engie for financial support. We also thank the Pró-Reitoria de Pesquisa e Pós-Graduação (PROPESP) from the Federal University of Pará (UFPA) (Notice February 2023).

**Institutional Review Board Statement:** All the methods herein applied were under the National Council for the Control of Animal Experimentation (CONCEA), SISBIO (Abio #1267/2020), and the Ethics Committee on Animal Use (CEUA-UFPA) guidelines.

**Informed Consent Statement:** Not applicable.

**Data Availability Statement:** The authors can be asked to provide the data.

**Acknowledgments:** The authors thank the field team, fishermen, pilots, and cooks for their assistance during the data-collection phase. The authors are also grateful to the Veterinary Hospital (HOVET) of the Universidade Federal Rural da Amazônia (UFRA) for providing the X-rays.

**Conflicts of Interest:** The authors declare no conflict of interest.

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
