# Peer review of "First Record of Spinal Deformity in the South American Silver Croaker Plagioscion squamosissimus (Eupercaria: Sciaenidae) in the Xingu River, Brazil"

_fishes, doi:10.3390/fishes8070363_

Round 1

Reviewer 1 Report

This manuscript reports the presence of vertebral deformities in wild-caught samples of  South American silver croaker Plagioscion squamosissimus. Following an extensive sampling (n=483 specimens) during a 12 month period, the authors discovered 9 specimens with severe axial deformations (1.9%). The manuscript provides a radiographic description of the one of these specimens, in comparison with the anatomy of a normal fish. The manuscript is of high interest since skeletal abnormalities may serve as indices of environmental disturbances. The manuscript merits publication, but only after some critical revisions are made. 

-        The introduction is mostly based on literature on salmonids. Since salmonids follow a significantly different ontogenetic pattern than the studied species, it should be strongly recommended to use more targeted literature; i.e. literature on the anatomy and ontogeny of vertebral abnormalities in species with a similar to P. squamosissimus ontogenetic pattern. Such an approach would assist authors on hypothesis forming on the critical ontogenetic period during which the recorded deformities developed (and thus causative factors acted).

-        The introduction should focus on the presence of skeletal deformities in the wild and their value as indices of environmental disturbances and pollution. This will strengthen the value of this manuscript. Currently, except of the salmon literature, the intro is largely based on two reviews on deformities in reared fish.

-        Materials and methods. When specimens were examined for deformities? Before or after fixation? What was the examination method (e.g., examination of the external morphology for the presence of morphological abnormalities)? Were all the anatomical parts examined (e.g, skull, fins and axis)?

-        Material and methods. Why only the 2 out of the 9 abnormal specimens were radiographed? Did all the 9 fish present the same gross abnormal phenotype (i.e., body shortening)?

-        Results. Fig. 2. Drawn lines do not allow the observation of the anatomical features. Please remove them and use insets of the areas of interest with larger magnification. To assist the reader to follow your counting, please indicate the 1st precaudal and caudal vertebra.

-        Results. Please also give the radiography of the second x-rayed abnormal specimen.

-        Discussion. Ontogenetic perspective might assist the authors to form some solid hypotheses on the critical period and thus on the possible causes of the observed deformities. e.g., which is the most possible stage for these deformities to develop. Could these deformities be the result of early notochordal abnormalities (e.g., Andrades et al 1996, Aquaculture 141, 1–11; Loizides et al. 2014, J Fish Dis, 37: 949-957); or did they develop on initially normally developed vertebral column, due to factors acting at later life stages (e.g., Printzi et al. 2021, J Fish Biol. 98: 987– 994; Johnson et al. 2020, Environ. Sci. Technol. 54, 5, 2892–2901)?

-        Discussion. A deeper discussion on some known causative factors of vertebral abnormalities would improve the manuscript and allow an appropriate hypothesis formation for future testing. For example, could factors like unfavorable temperature, heavy metal pollution or low oxygen levels during critical periods induce the observed abnormalities? Existing discussion is rather general and uses inaccurate literature (e.g. 15, line 129; 3, line 133). I would strongly recommend using more targeted literature, such as Sawada et al. (2018 Aquac Res. 49: 3176– 3186). This latter paper for example reports similar vertebral defects (fusions) to originate from the early embryo exposure to hypoxia/hypercapnia. 

Other comments

A number of comments is also made in the pdf of the ms.

Reviewer 2 Report

Review of Montag et al., fishes-2416431, “First record of spinal deformity in the South American silver croaker Plagioscion squamosissimus (Eupercaria: Sciaenidae) in the Xingu River, Brazil

1.      This is a short, simple and concise Brief Report. The writing is very clear and easy to follow; it appears to be free of errors. I have only a few simple comments and recommendations concerning this manuscript.

2.      In the Abstract (and elsewhere in the manuscript), the authors describe the main effect of the spinal deformity as “body shortening,” but it is difficult to determine how much actual shortening has occurred. Table 1 shows the body length of the deformed fish that were collected, but the reader cannot determine if this represents a real shortening of the body relative to fish without this spinal deformity. I recommend that the authors include, for purposes of comparison, body lengths of undeformed fish of similar mass. Or, perhaps the authors can describe the percentage decrease in body length caused by this spinal deformity (relative to the mean length of normal fish).

3.      The photos of fish are helpful, but it is interesting that the two fish differ notably in color. Could the authors include more information about body coloration in this species? Is there normally a lot of variation, or does color change with age/size, location, or time of year? Also, did the fish with the spinal deformity show any other differences from normal fish—not just color, but anything else?

4.      The Xingu River is very long. I am glad that the authors provided precise geographic coordinates and also a description of the section of the river where the study was performed, but I recommend that the authors add a simple figure showing the location of this river in Brazil with a mark where the study occurred. This will help readers to know exactly where the deformed fish came from (and where the dam is located).

5.      The Discussion is effective and the list of cited references is appropriate. However, I wish more information were included about deformities caused by injuries such as predation attempts or mechanical injury (fish being struck, or perhaps swimming into an obstruction). Do fish spinal deformities occur mainly when individuals are harmed as younger fish that are still developing and growing a lot, or can injuries to adults also cause spinal changes that lead to body shortening? I am sure there is more information about this and more sources to be cited. [Of course, I understand that there are other causes of deformity including genetic defects and environmental problems.]

6.      I thank the authors for ensuring that their submission was well written and carefully edited and proofread.

Round 2

Reviewer 1 Report

The authors followed many or my suggestions, but not those related to the introduction and the discussion which continue been week.

Introduction. It needs to target on the deformities in the wild and not in aquaculture. In this way, the introduction will succesfully adderss the significance of recording these deformities in wild stocks. Also, it should be interesting to refer to why generally (e.g., climate change, pollutants, water acidification) deformity rates increase in the wild. Some suggestions here with appropriate literature could be important. Also, it should be interesting to mention that morphological abnormalities are used as indices of environmental disturbances in the wild. 

Lines 39-40. "As a result ..." of what ? The previous sentence referes to the causative factors of skeletal defects. Also, I do not agree with the statement on the "most common cases are vertebral abnormalities such as flattening or 39 fusion of vertebrae, causing curvature and/or shortening of the spine". This could be the case of reared salmonids, but not of the rest fish (wild mostly, but also reared).

Line 47. Used literature is irrelevant (a general review). Line 50-51. The connection between hydroelecric dams and elevated deformities is largely speculative and not clear. 

L52-L53. Not really scarce (see your next sentence, citating a different review). Used literature is not appropriate. What is the relevance of this scarsity (and part L52-L56) with the present manuscript?

L111. Based on which terminology? Your x-rays just show an urostyle. 

L143. Why in contrast? Do you mean that they are smaller than the normal? Why is this difference is attributed to growth and not to a different age of the abnormal specimens?

L189-L191. It should be noted that vertebrae were counted on the basis of their processes (neural or haemal) and not of their centra, which were not possible to be counted beacuse of the extensive fusions.

L192-L194. Highly speculative hypothesis. In a variety of species, swimming exercise induces haemal lordosis and not vertebral fusions. Please see suggested literature in my previous review on some potential factors of vertebral fusions. 

L192-L194. Why is larval stage targeted? Why not the later stages? You need literature to support that (e.g., connecting notochord abnormalities with later vertebral defects).

L201-203. Partial repetition of introduction. Discussion whould be more specific. L206-207, similarly is a partial repetition of the previous discussion parts.

L213-L214. For sure not in farmed fish!! Inaccurate and general literature.

Discussion. Needs substantial improvements, avoidance of repetitions and targeting on potential causes of vertenral fusions and compressions (see my previous review). Not necessary to be big, but it need to be focused.

Consclusions need also to suggest what abiotic parameters should be monitored in parallel with the abnormalities. 
